# 30 Year Trends of Reduced Physical Fitness in Undergraduate Students Studying Human Movement

**DOI:** 10.3390/ijerph192114099

**Published:** 2022-10-28

**Authors:** Myles W. O’Brien, Madeline E. Shivgulam, William R. Wojcik, Brittany A. Barron, Roxanne E. Seaman, Jonathon R. Fowles

**Affiliations:** 1School of Kinesiology, Acadia University, Wolfville, NS B4P 2R6, Canada; 2Division of Kinesiology, Faculty of Health, School of Health and Human Performance, Dalhousie University, Halifax, NS B3H 4R2, Canada

**Keywords:** university students, kinesiology, physical education, physical fitness, objectively measured physical activity

## Abstract

The physical fitness of Canadian adults has decreased over the past 30 years, while sedentary time has increased. However, it is unknown if university students studying human movement exhibit similar population declines. Physical fitness (i.e., anthropometrics, musculoskeletal fitness, and aerobic fitness) and habitual activity (e.g., physical activity, stationary time, etc.) was measured in a cohort of kinesiology students (Post30; 2010–2016; *n* = 129 males, 224 females) using standardized fitness testing and accelerometry, respectively. Physical fitness was compared to data collected on a cohort of students from the same institution and program, 30 years prior (i.e., Pre, 1984–1987, *n* = 103 males, 73 females). Post30 had greater waist circumference (males: 83.6 ± 9.1 cm vs. 77.8 ± 8.3 cm, females: 77.1 ± 9.7 cm vs. 70.3 ± 5.2 cm, both *p* < 0.001) and lower estimated maximal aerobic fitness (males: 55 ± 11 vs. 63 ± 7, females: 45 ± 10 vs. 50 ± 7 mLO_2_/kg/min, both, *p* < 0.001). Compared to Pre, male Post30 vertical jump (53.6 ± 10.2 vs. 57.0 ± 8.4 cm, *p* = 0.04) and female Post30 broad jump (178.7 ± 22.1 vs. 186.0 ± 15.5 cm, *p* < 0.001) were lower. A subsample (*n* = 65) of Post30 whose habitual activity was assessed, met the aerobic portion of Canadian physical activity guidelines (~400 min/week), but spent excessive amounts of time stationary (10.7 h/day). Current kinesiology students may not be immune to population decreases in physical fitness. Relative to previous group of students interested in movement, fitness was lower in our sample, potentially attributed to excessive stationary time. Regular assessment of physical fitness in kinesiology curriculums may be valuable to understand these declining trends in undergraduate students that mimic population declines in fitness.

## 1. Introduction

The health benefits of being physically active are well established [1], with additional benefits achieved through higher levels of physical fitness [2]. Canadians have increased in bodyweight over the past three decades, with rates expected to continue to increase over the next two decades [3,4]. Compared to their age-matched peers 30 years prior, Canadian males and females aged 20–39 during the years 2007–2009 had a higher body mass index (BMI), waist circumference, and skinfold thickness but lower muscular strength and flexibility [5]. The Canadian populations’ decrease in fitness has been attributed, at least in part, to decreased physical activity and increased sedentary behaviors [6].

A large proportion of post-secondary students are excessively physically inactive [7,8] and importantly, sedentary time has increased over the last 10-year period among university students [9]. However, university students enrolled in kinesiology programs are more aerobically fit and self-report more activity than non-kinesiology majors [10]. It is unknown if a similar trend of declining physical fitness is evident among students interested in the study of movement and well-versed in the health benefits of higher aerobic fitness. One may presume that because of the chosen field of study and greater activity level [10], students whose education is grounded in movement would be more likely to maintain higher fitness and may be immune to the declining fitness trends exhibited by the general population.

Physical activity is any bodily movement produced by skeletal muscle that results in energy expenditure while physical fitness is a set of attributes that are either health- or skill-related that can be measured with specific tests [11]. Physical fitness encompasses a series of components including morphological fitness (e.g., body mass index, waist circumference) muscular fitness (e.g., strength, muscular endurance, flexibility), motor fitness (e.g., speed, agility), cardiorespiratory fitness (e.g., aerobic fitness), and metabolic fitness (e.g., blood lipid profile, glucose tolerance, insulin sensitivity) [12]. Accordingly, a battery of physical fitness tests may provide insight into specific aspects of fitness that vary between or within-groups.

We sought to characterize kinesiology students in a recent cohort (from 2010–2016) and compare them to students from the same school and similar program from 30 years prior (data collected in regular testing from 1984–1987). These comparative cohorts are from the same institution that has retained most aspects of what was previously a ‘physical education’ program into the, now named, ‘Bachelor of Kinesiology’ program. It was hypothesized that among the unique sample of activity-minded individuals, fitness levels of current kinesiology students would not be different than the physical education students 30 years prior. Due to the well-established impact of sex on physical fitness [13,14], analyses were conducted separately for males and females.

## 2. Materials and Methods

### 2.1. Participants

Participant testing and secondary use of previously collected data was approved by the institution’s Research Ethics Board (REB 15-61). Fitness testing results were obtained from the university’s Bachelor of Physical Education students of 1984–1987 (Pre; *n* = 178), which were collected as part of the past degree’s mandatory fitness testing. Kinesiology students from 2010, 2015, and 2016 (Post30, *n* = 348) were recruited through a first-year Bachelor of Kinesiology compulsory course. Both cohorts of participants were collected from the same university in Atlantic Canada. This reference base is an appropriate cohort for comparison of students interested in the study of human movement, as the current Bachelor of Kinesiology program has preserved much of the practical and applied roots of the previous Physical Education program where many past and present students go on to teach physical education or pursue occupations in applied kinesiology and health professions.

The pooled sample difference for waist circumference between the 2007–2009 Canadian Health Measures Survey with the 1981 Canada Fitness Survey was used to estimate a sample size for this study [5]. Based on the moderate-large effect size (effect size = 0.61), a sample size calculation estimated that a minimum of 44 participants per group were needed assuming a two-tailed, α = 0.05 and β = 80% power (G*Power, v3, Heinrich Heine Universitat, Dusseldorf, Germany) [15].

### 2.2. Physical Fitness Assessments

Post30 fitness assessment protocols were selected to replicate the testing done at Pre. In the mid 80s, the measurement protocols for assessing anthropometrics, musculoskeletal fitness and aerobic fitness were from protocols available at the time [16,17,18]. The fitness tests and measures were conducted by Canadian Society for Exercise Physiology (formerly Canadian Association of Sport Sciences) fitness appraisers. Prior to testing, all students refrained from large meals, alcohol, caffeine, nicotine, vigorous exercise, and blood donations. The order of fitness assessments were randomized over two sessions with plenty of rest-time (~15 min) between assessments. The 12-min run was conducted during a dedicated third visit. The exercise protocols were identical in subsequent ‘post’ testing, except Post30 vertical jump height was measured using a Vertec (Sports Imports, Columbus, OH, USA), while Pre vertical jump was measured against a wall using the Sargent Jump Test protocol [19]. Peak vertical jump power was calculated using the Sayers equation: Power (W) = [60.7 × vertical jump displacement (cm)] + [45.3 × weight (kg)] − 2055 [20].

The anthropometric measures collected included height (cm), weight (kg), and waist girth (cm) using standard anthropometric protocol [16]. BMI was calculated as body weight/(height)^2^. Muscular fitness was assessed through vertical jump, broad jump, sit-and-reach, and grip strength using standard techniques [16,17,19]. The vertical and broad jumps were performed in a stationary position with an arm swing and use of the stretch shortening reflex permitted. Two-foot broad jump was set up with a tape measure running down the left side of the jump area, with distance recorded at the heel of their back foot. Flexibility was assessed using the standard sit-and-reach test, in which participants asked to reach as far as possible without bending the knees as they sat on a floor mat with shoes removed and their legs extended against a sit-and-reach flexometer [17]. Grip strength was assessed using a calibrated hand-grip dynamometer on each hand. The maximum scores from each hand were combined for their overall score. The Illinois Agility test followed an outlined standardized procedure [21]. A 50-m sprint was conducted, and sprint time was measured by two testers using handheld stopwatches at the first movement to the chest crossing the finish line. Sprint times were averaged to determine participant scores. PE80s grip strength and 50 m sprint was not completed in 1984, but was in 1985–1987, while waist circumference and vertical jump was only completed in 1986/1987.

Aerobic fitness was measured using the Cooper 12-min run on a 400-m track [22]. This track had a rubberized surface for Post30, but the surface for Pre30 is unclear. A rubberized surface may be an advantage for the Post30, but we believe this to be minor given our observed results. Participants ran around the track as many times as possible in 12 min; distance was rounded to the nearest 100 m and recorded. Participants predicted maximal aerobic capacity (VO_2_max) was calculated based on the distance ran using the following equation: Predicted VO_2_max (mL/kg/min) = (Distance ran (m) × 0.0268) − 11.3 [23].

### 2.3. Habitual Activity in a Sub-Sample of Current Students

A subsample of kinesiology students from 2015 and 2016 (*n* = 65; 25 males) completed a Modified-Leisure Time Physical Activity Questionnaire (mLTPA-Q) [24] and wore a validated [25,26] physical activity monitor (PiezoRx^®^, Steps Count Inc., Deep River, ON, USA) for one full week.

Students instructed to wear the device a minimum of 10 h per day for one full week. PiezoRx monitors were selected due to their ability to measure steps, light-, (LPA), moderate- (MPA), and vigorous-intensity physical activity (VPA), total physical activity (TPA), and the number of 10-min bouts. Stationary time was calculated by subtracting TPA from wear time and represents the combination of sedentary time and standing time, which cannot be distinguished using waist-worn or wrist-worn accelerometers. The PiezoRx has been previously validated in controlled laboratory [26] and free-living [25] conditions and determines time spent in activity intensity based on step rate thresholds adjusted for height [27]. A minimum of 4-days of wear time was required for inclusion, consistent with recommendations [28]. The mLTPA-Q was used to self-report the number of days per week they participate in planned aerobic exercise and resistance exercises [24].

### 2.4. Statistical Analysis

Linear regression analysis was implemented within each sex to compare Pre and Post30. Pre/Post30 were inserted as the independent variable (i.e., 0/1), and the physical fitness measure as the outcome variable. The unstandardized-β represented the mean difference between Pre and Post30, and statistical significance of this predictor variable indicated a group difference. Cohen’s *d* effects sizes were calculated between Pre and Post30 within each sex as (Pre − Post30) ÷ pooled standard deviation.

Normality was assessed via the Kolmogorov–Smirnov test within each sex. For both sexes, age, BMI, sit and reach, and the agility tests were non-normal (all, *p* < 0.001). For males, waist circumference and sprint times were non-normal (all, *p* < 0.002). For females, grip strength was non-normal (*p* < 0.001). While thresholds for skewness vary, median values with ranges are presented for outcome variables exceeding ±2 (i.e., age for both sexes and 50-m sprint for females only; both, >3.9). Independent sample *t*-tests evaluated physical fitness differences in the sub-sample of 2015/2016 students who wore the PiezoRx^®^ pedometers to the entire Post30 sample and confirmed that the sub-samples fitness was representative of the entire group (all, *p* > 0.44). The sub-sample of participants with physical activity data were analyzed using regressions, with sex incorporated as the dichotomous predictor variable. Analysis was completed in SPSS Statistics (Version 27.0, IBM, New York, NY, USA). Statistical significance was defined as *p* < 0.05. All physical fitness and habitual activity variables were described using mean ± standard deviation.

## 3. Results

### 3.1. Characterization of Kinesiology and Physical Education Students’ Physical Fitness

Participant physical fitness characteristics are presented in Table 1. Pre females (*n* = 75) and males (*n* = 103) had mean ages of 19.4 ± 3.0 (median, range: 18, 18–36) years and 19.4 ± 2.1 (19, 18–30) years, respectively. The Post30 cohort were primarily females (*n* = 219) with a mean age of 19.0 ± 2.0 years (18, 18–35), while males (*n* = 129) had a mean age of 19.3 ± 2.0 (19, 18–35) years. Female Post30 weighed more than female at Pre (66.3 vs. 62.5 kg, *p* = 0.01; Table 1), resulting in a greater BMI (24.0 vs. 22.4 kg/m^2^, *p* = 0.004). Both male and female Post30 had larger waist circumferences (male = +5.8 cm; female = +6.8 cm; both, *p* < 0.001) compared to Pre of the same sex.

Compared to Pre, Post30 males had a 3.4 cm lower vertical jump height and Post30 females had a 7.3 cm lower broad jump (both, *p* < 0.045). No differences between cohorts were observed for vertical jump power, sit and reach, Illinois agility test and 50 m sprint (all, *p* > 0.12). The median and range for 50 m sprint for Pre and Post30 females was 8.2 (6.4–11.9) and 8.1 s (7.0–13.4), respectively. Post30 male’s grip strength was 12.3 kg lower compared to Pre (*p* < 0.001). Predicted VO_2_max was lower in Post30 males (63 vs. 55 mL/kg/min), and females (50 vs. 45 mL/kg/min) compared to the Pre groups (both, *p* < 0.001; Figure 1).

### 3.2. Habitual Activity among Sub-Sample of Kinesiology Students

Objectively measured PA of Post30 was not different between sexes, as shown in Table 2 (all, *p* > 0.60). Most activity was of moderate-vigorous-intensity (76% of total activity), with less time spent in LPA (24%). There were no significant sex differences for the number of resistance training sessions per week (2.9 ± 2.2 vs. 1.9 ± 1.8, *p* = 0.06), or planned aerobic exercise sessions per week (2.6 ± 2.1 vs. 3.4 ± 2.1; *p* = 0.13).

## 4. Discussion

The purpose of this study was to characterize physical fitness levels of kinesiology students in comparison to an analogous cohort of students interested in the study of movement from 30 years prior that attended the same institution. We demonstrated that a recent cohort of kinesiology students are less aerobically physically fit and have a greater waist circumference than a similar cohort of students ~30 years ago. Despite both cohorts of students studying human movement and being educated on the multitude of benefits of physical fitness, students in our sample had increased adiposity and lower fitness, similar to the trend observed in the greater Canadian population [5].

Waist circumference estimates abdominal obesity and has implications on obesity-related comorbidities, independent of BMI [29]. Post30 had a waist circumference that was 6–7 cm larger than Pre (or 7–9% higher), which in the absence of performing better on muscular fitness tests, supports the presence of a greater abdominal adiposity and an unhealthier physical phenotype [30]. For example, Post30 females weighed ~3.8 kg more than Pre females and broad jumping performance was lower in Post30 (Table 1). Based on heuristic thresholds for Cohen’s *d,* although without limitations [31], the differences in waist circumference among males and females achieved a moderate-strong effect size (between 0.5–0.8), accordingly we position that a 6–7 cm higher waist circumference is practically meaningful. Waist circumference differences were also observed when comparing the 2007–2009 Canadian Health Measures Survey with the 1981 Canada Fitness Survey that showed 21st century adults’ waist circumference increased from 85 cm to 91 cm for males and 72 cm to 83 cm for females [5]. Our study observed higher waist circumference values in our group of physical activity minded individuals, rather than the general public. In general, the physical fitness values of our participants across most metrics were in the “good” to “excellent” categories of normative values [32]. While the sample of kinesiology students included herein may still perform well in comparison to the general population, there is a clear trend of increased adiposity relative to a comparison group of students 30 years prior.

Aerobic fitness is a strong predictor of morbidity and mortality in the population across a range of conditions [33,34]. According to the current CSEP health benefit ratings for predicted VO_2_max [32] our sample at Post30 were, on average, in the “very good” category and Pre students were in the “excellent” category. This suggests that Bachelor of Kinesiology students in our study were still aerobically fit; albeit less than a similar group of movement-focused students from the 1980s. This difference in aerobic fitness (~6–9 mL/kg/min or ~2 metabolic equivalents) exceeds that of the Canadian public of who declined in aerobic fitness by ~1–6 mL/kg/min over a similar 30-year time-period [35]. For context, a 1 metabolic equivalent (multiple of resting oxygen consumption) reduction in aerobic fitness may confer a 12–35% decrease in survival risk [36]. Even within a decade (2007 to 2017), Canadian men and women have demonstrated a ~2–3 mL/kg/min decline in aerobic fitness, but a decreased sit and reach performance among males [37], whereas flexibility was similar between groups in our study. While aerobic fitness is influenced by body composition, sex, and genetics, it is also heavily impacted by activity-related factors (e.g., exercise) [38,39]. While speculative, it may be possible that previous students engaged in more frequent or intense aerobic training than the sub-sample of current kinesiology students, who engaged in 2–3 planned aerobic and 2–3 planned resistance exercise sessions weekly. Aerobic and resistance exercise participation and the composition of training sessions (i.e., frequency, intensity, type, time) of the previous cohort is unknown and may contribute to between-group differences observed. Alternatively, stationary time has been demonstrated to be inversely associated with maximal aerobic fitness in the Framington Heart Study [40] and in healthy adults [41]. While the sub-sample of current students reported a lot of MVPA, little time was spent engaging in LPA and a lot of time spent in stationary postures (Table 2). It is plausible that the Post30 students included spend more of their leisure-time engaged in screen time (e.g., videogames, computers, etc.) than was available in the mid-1980s, which results in more time stationary, and therefore a lower aerobic fitness [42]. Certainly, this theory is hypothesis-generating given the inability to objectively measure habitual activity from the mid 1980s, but it is plausible given the high amount of stationary time in our study’s current students. In American adults, the proportion of adults engaged in exercise, sport, or lifestyle physical activity in general has decreased from 1988 to 2017 [43], whether this impacts students who are interested in movement warrants further study. While the physical habits of university students are often overlooked, more regular physical fitness testing may be warranted in university settings to better determine whether a trend of declining fitness is observed among physical-activity conscious students. This coincides with the studies of current students who are required to take health promotion, exercise physiology, and/or fitness assessment type courses.

This study adds to the current literature by evaluating physical fitness of a sample of university kinesiology students with a similar cohort of university students 25–30 years ago. Even among students who are knowledgeable on physical fitness, the Post30 kinesiology students in our study exhibited larger waist circumferences, poorer muscle power/strength, and worse predicted aerobic fitness. Our kinesiology students appear to follow the same declining health patterns of the public, highlighting the impact of modern lifestyles (e.g., fast food, screen time, etc.). The implementation of more frequent fitness testing in current curriculum may draw attention to what may be a growing problem. Regular fitness testing was common practice many years ago and may help to identify trends and promote healthier outcomes for current and future students. Students interested in movement may benefit most from the practical experience gained by having their fitness assessed and/or senior students being the fitness appraisers. The integration of this testing may prove a useful educational experience for this specific sample. As well, the amount of time spent sedentary and in vigorous activity is likely important in the degree of adiposity and aerobic fitness. Tracking physical activity and sleep levels (i.e., 24-h movement guidelines) may be a learning opportunity for those students interested in movement to broaden their perspectives of the influence of physical activity on health to include sedentary behaviour, light physical activity, and vigorous physical activity (i.e., sports) aside from structured exercise.

## 5. Limitations

The recent students are from the same academic program, that was previously a Physical Education program but is now a Bachelor of Kinesiology, that has maintained a similar focus over the years on a balance of content in both the science of movement as well as teaching and coaching and theory and practical and applied contexts (as opposed to a Bachelor of Science in Kinesiology that may focus purely on science aspects and cater to a different type of student). It is possible that differences may even be larger among science focused programs that focus less on professional or applied contexts whose curriculum more so emphasizes graduate or medical school preparation. As the specific composition of each program was not ascertained and is therefore a limitation of this observational analysis, the findings are still interesting and noteworthy.

Our comparison of kinesiology students to similar students 30 years apart requires the assumption that students interested in either of these programs are interested in ‘movement’ to a similar amount. However, this cannot be assessed. Although objective activity was measured in current kinesiology students, there is no way to determine the activity level of past physical education students, so it is possible that the difference in fitness between the cohorts may be due to differences in the characteristics of the cohorts themselves. Many current kinesiology students are involved in varsity sports, recreationally competitive sports, and intramural sports, as was the assertion for physical education students of yesteryear, but without objective data on sport participation this is impossible to ascertain. The test–retest reliability may be influenced by different fitness appraisers between cohorts; however, this discrepancy is believed to be minimized by using senior exercise appraisal students supervised by trained exercise appraisers in both cohorts. Strong efforts were made to complete protocols according to published methods and standard protocols between the cohorts; however, no two testers were present for both testing cohorts 30 years apart.

## 6. Conclusions

Despite having relatively high fitness levels, this sample of current students who are interested in the study of movement have more abdominal adiposity and are less aerobically fit than a fellow sample of students educated in human movements from the 1980s. Those educated on the importance of movement may not be immune to the trends of decreasing fitness experienced by the rest of the Canadian population, possibly due to spending too much time stationary.

## Figures and Tables

**Figure 1 ijerph-19-14099-f001:**
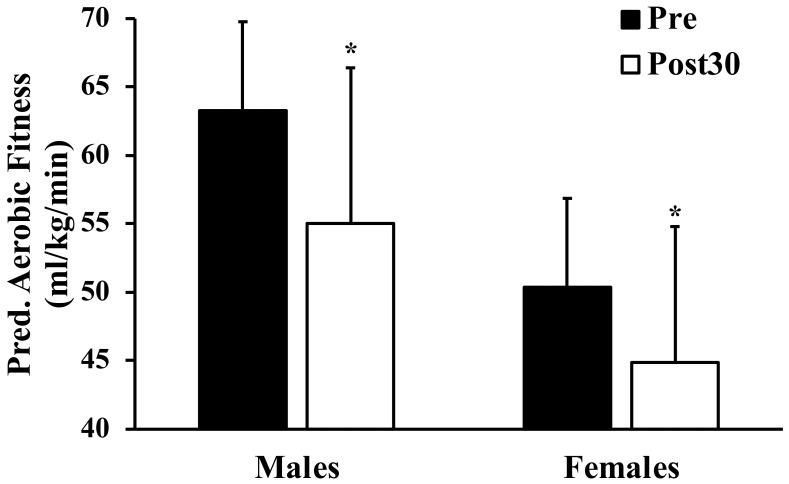
Comparison of predicted VO_2_max assessed via Cooper 12-min run between Pre (current students) and Post30 (students from 30 years ago), * *p* < 0.001 between Pre and Post30 of same sex.

**Table 1 ijerph-19-14099-t001:** Anthropometric and physical fitness comparison of undergrade students measured 30 years apart.

	Males					Females				
Variable	*n*	Pre	*n*	Post30	Cohen’s *d*	*n*	Pre	*n*	Post30	Cohen’s *d*
Height (cm)	101	178.6 ± 6.0	129	178.4 ± 7.1	0.03	74	166.1 ± 6.5	219	166.3 ± 7.1	−0.03
Weight (kg)	102	78.0 ± 10.0	129	80.4 ± 13.8	−0.20	74	62.5 ± 9.2	219	66.3 ± 9.6 *	−0.40
BMI (kg/m^2^)	101	24.5 ± 2.9	129	25.2 ± 3.6	−0.21	74	22.4 ± 2.6	219	24.0 ± 4.0 *	−0.43
Waist Cir. (cm)	44	77.8 ± 8.3	126	83.6 ± 9.1 *	−0.65	36	70.3 ± 5.2	215	77.1 ± 9.7 *	−0.74
Broad Jump (cm)	102	232.2 ± 17.2	130	226.4 ± 31.3	0.22	72	186.0 ± 15.5	204	178.7 ± 22.1 *	0.35
Vertical Jump (cm)	46	57.0 ± 8.4	132	53.6 ± 10.2 *	0.35	39	40.1 ± 7.9	204	37.9 ± 9.9	0.23
Vertical Power (W)	46	4824 ± 640	132	4859 ± 882	−0.04	38	3239 ± 642	197	3243 ± 816	−0.01
50 m Sprint (s)	74	6.9 ± 0.5	122	7.0 ± 0.6	−0.18	51	8.4 ± 1.5	189	8.3 ± 0.8	0.10
Illinois (s)	101	17.1 ± 0.8	117	17.3 ± 1.4	−0.17	73	19.1 ± 1.6	189	19.3 ± 1.7	−0.12
Grip Strength (kg)	80	103.4 ± 18.7	123	91.2 ± 22.5 *	0.58	57	64.9 ± 19.8	220	62.6 ± 14.3	0.15
Sit-and-Reach (cm)	100	28.4 ± 15.7	125	30.6 ± 9.0	−0.18	72	32.0 ± 18.0	223	33.5 ± 10.7	−0.12
12-Minute Run Distance (m)	101	2784 ± 243	118	2476 ± 428 *	0.87	71	2303 ± 243	194	2096 ± 368 *	0.61

Note: Pre, physical education students from 1984–1987; Post30, kinesiology students from 2010, 2015, & 2016. BMI, body mass index; Cir., Circumference; ** p* < 0.05 to Pre of same sex. Cohen’s *d* was calculated as (Pre − Post30) ÷ Pooled SD. Data presented as mean ± SD.

**Table 2 ijerph-19-14099-t002:** Objectively measured physical activity in current kinesiology students.

Variable	Males (*n* = 25)	Females (*n* = 40)
Daily Steps	8657 ± 2437	8472 ± 3265
Light-Intensity Activity (min/week)	127 ± 102	124 ± 75
Moderate-Vigorous-Intensity Activity (min/week)	412 ± 111	385 ± 145
Moderate-Intensity Activity (min/week)	360 ± 88	334 ± 112
Vigorous-Intensity Activity (min/week)	53± 31	51 ± 64
Total Physical Activity (min/week)	540 ± 170	509 ± 192
>10 min Activity (bouts/week)	8.5 ± 6.7	6.4 ± 6.9
Stationary Time (h/day)	10.7 ± 0.4	10.8 ± 0.5

Note: No sex differences as assessed via regression analysis (sex entered as a predictor variable and each activity variable as an outcome) were observed (all, *p* > 0.50). Data presented as mean ± SD.

## Data Availability

All data files can be provided by the corresponding author upon reasonable request.

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
