# Peer review of "30 Year Trends of Reduced Physical Fitness in Undergraduate Students Studying Human Movement"

_ijerph, 2022, doi:10.3390/ijerph192114099_

Round 1
Reviewer 1 Report
Dear authors,
Your submitted paper "30 Year Trends of Reduced Physical Fitness in Under-graduate Students Studying Human Movement” is beneficial for comparing the level between the students during the long time period.
However, I recommend making some minor corrections before issuing this paper.
Abstract
Row 28 – females: 77.1±9.7 cm vs 83.6±9.1 cm – I think that correct is 70.3 ± 5.2, not 83.6±9.1 cm
Introduction
The introduction is brief and related only to the assumption of a decreasing level of the common generation of students. Your recommendation, in conclusion, is also aimed at the population; therefore, I recommend adding one paragraph that mentions whether the selected variables in this study sufficiently represent the physical level of the population.
Although you stated during the introduction that a sedentary life could decrease the level of physical fitness level thus the hypothesis aimed that the physical fitness level of the pre and post30 students would not be different. Why?
Methods
You should mention the surface where were performed the tests related to running. Was the same for pre and post30 participants?
Results
Row 180 - „Female Post30 weighed more than female at Pre (62.5 vs. 66.3 179 kg, P=0.01; Table 1)“ – There is stated the first Post30 and then Pre, but I think that the results are reversed, the first 62.5 vs 66.3?
Reviewer 2 Report
I would like to thank the Authors for the opportunity to review the article submitted to the International Journal of Environmental Research and Public Health. This is indeed an unprecedented time and such a study is needed to navigate through the menace of this time. However, in my opinion, there are some issues that need further clarification prior to considering the paper for publication. Please refer to my specific comments below:
1. In relation to the sample size it is necessary to present the sampling error, the margin of error, as well as the confidence level.
2. In the case of independent variable, you should use descriptive statistics as well as Levene's test and the test to check the normality of the distribution. Not only the results of, for example, the Kolmogorov-Smirnov test and the Levene's test, but also the value of skewness and kurtosis should be added.
3. The analysis result is too simplistic. A more sophisticated analysis is needed. Why do not you calculate effect size for the t-tests? In addition, the regression analysis or structural equation analysis is recommended.
4. Besides, you should add the limitations of study section as a sub-section under or below the conclusion.
5. There is no in-depth literature review. The references are mostly from the old days. The references are rather dated, except for a few in 2019 and 2022. The reviewer would like to see comparisons in the discussion section with more updated peer reviewed journal papers, such as in the years of 2019 and 2022. I believe that the bibliographic references are insufficient and additional research should be reviewed, especially in the discussion section.
6. In relation to the level of statistical significance you should improve ‘P’ to ‘p’. You should use a lowercase letter and the italics.
7. What is more, the generalizations should be avoided. Unfortunately, the generalizations appears especially in the discussion section.
8. It would be appreciated if the authors could give more details about practical implications of the study.
I am pleased to wait for your revised version then.
Best regards.
Round 2
Reviewer 2 Report
Great job!
Kind regards.